# In-Context Demonstration Selection with Cross Entropy Difference

**Dan Iter, Reid Pryzant, Ruochen Xu, Shuohang Wang,**
**Yang Liu, Yichong Xu, Chenguang Zhu**
Microsoft Cognitive Service Research
`iterdan@microsoft.com`

## Abstract

Large language models (LLMs) can use in-context demonstrations to improve performance on zero-shot tasks. However, selecting the best in-context examples is challenging because model performance can vary widely depending on the selected examples. We present a cross-entropy difference (CED) method for selecting in-context demonstrations. Our method is based on the observation that the effectiveness of in-context demonstrations negatively correlates with the perplexity of the test example by a language model that was finetuned on that demonstration. We utilize parameter efficient finetuning to train small models on training data that are used for computing the cross-entropy difference between a test example and every candidate in-context demonstration. This metric is used to rank and select in-context demonstrations independently for each test input. We evaluate our method on a mix-domain dataset that combines 8 benchmarks, representing 4 text generation tasks, showing that CED for in-context demonstration selection can improve performance for a variety of LLMs over baseline selection methods.[1]

## 1 Introduction

Large language models (LLMs) have been widely successful across many NLP tasks (Bommasani et al., 2022; OpenAI, 2023; Bubeck et al., 2023). The primary method for LLMs to adapt to new tasks has been using in-context learning, where a few examples and labels are provided as input to the model (Agrawal et al., 2022). This simple approach has shown large improvements over zero shot settings, and even outperformed finetuning methods when the training dataset is small. However, the model's performance can be greatly influenced by which in-context demonstrations (ICDs) are selected into the prompt (Lu et al., 2022; Zhao et al., 2021a; Min et al., 2022).

Selecting the best in-context demonstrations can be challenging. The variance in performance of similar demonstrations can be large, and the selected examples can introduce unfavorable prior biases on the output label space (Zhao et al., 2021b). The naive approach is to randomly sample demonstrations from the same source dataset. Previous methods for selecting ICDs include simple methods such as selecting nearest neighbors by embedding distance (Liu et al., 2022b) and retrieval-based methods that require training a retriever model (Rubin et al., 2022). This work presents a new method of selecting demonstrations that can be applied to any sized training data, requires training small PEFT models only and outperforms the nearest neighbor baseline on GPT-Davinci-003 (Ouyang et al., 2022).

We propose a cross entropy difference (CED) method for ICD selection. CED has been used to select in-domain data from large mixed-domain datasets for domain adaptation (Axelrod et al., 2011; Moore and Lewis, 2010; Wang et al., 2018). We borrow this idea to conduct ICD selection.

Specifically, we utilize parameter efficient finetuning to train small models on training data that are used for computing the CED between a test example and every candidate in-context demonstration. The CED scores are used to rank and select in-context demonstrations. We present a theoretical explanation for the effectiveness of CED. CED approximates the gradient alignment between training and test examples. Our analysis builds on previous findings that demonstrations operate as "meta-gradients" and shows that demonstrations with gradients similar to those of test inputs lead to improved performance in downstream tasks (Dai et al., 2022).

We evaluate our proposed CED-ICD selection method on a mixed-domain dataset composed of 8 datasets on 4 tasks: binary classification, multiple choice and extractive question answering and ab-

---

[1]Code is available at https://github.com/microsoft/LMOps

stractive question answering. We show that down-stream model performance using CED-ICD out-performs nearest neighbor baselines and transfers across models allowing training small models for selection but evaluating test examples on much larger models including GPT-3.5.

The contributions of this work are the following: 1) We present a method for selecting in-context demonstrations based on cross entropy difference. 2) We provide theoretical guidance for why selecting demonstrations based on their gradient alignment with test example is an effective heuristic. 3) We evaluate our method on a mixed-domain benchmark composed of 8 datasets from 4 tasks. 4) We evaluate our method on different sizes of GPT3 models, showing that this method transfers to larger models leading to performance improvements over baseline selection methods.

## 2 Related Work

Our work combines ideas from three bodies of research: In-context learning (ICL), data selection for domain adaptation and parameter efficient finetuning (PEFT).

While in-context learning (Agrawal et al., 2022) has shown very strong results for few-shot settings, recent work has shown that LLMs are very sensitive to the selected examples leading to large variance in performance (Zhao et al., 2021b), sensitivity to the order of examples (Lu et al., 2022) and even lack of sensitivity to the actual labels (Min et al., 2022). Other work has attempted to mitigate these challenges by selecting in-context demonstrations by the nearest neighbor examples in embedding space (Liu et al., 2022b), or training a retrieval mechanism (Rubin et al., 2022). We build on this line of work by proposing a novel selection method that combines the observations from Min et al. (2022) that domain similarity is a key characteristic of good in-context demonstrations, and observations from Gonen et al. (2022) that using perplexity could be a good heuristic for prompt selection.

Previous work on domain adaptation has focused on finding in-domain examples from a large out-of-domain dataset to train a model that achieves a better generalization on a target distribution (Moore and Lewis, 2010; Axelrod et al., 2011; Grangier and Iter, 2022). Data selection is intended to maximize the distributional similarity between a training dataset and test dataset. However, cross entropy dif-

ference has not been used previously at the example granularity to rank the "in-domainness" of training data in reference to just a single target example. We propose a natural extension of this framework for selecting demonstrations that are "in-domain" for a test input, which we demonstrate is an effective metric for selecting demonstrations for in-context learning.

Parameter efficient finetuning (PEFT) proposes a class of methods for augmenting model parameters with a small number of additional parameters that can be trained and stored efficiently (Lester et al., 2021; Li and Liang, 2021; Liu et al., 2022a; Hu et al., 2022). However, PEFT is usually used independently from in-context learning. Liu et al. (2022a) report that combining in-context learning and PEFT has not been effective. Sun et al. (2023) does report some settings where PEFT and ICL can be combined, but only under specific task conditions. We report similar findings, that in-context demonstrations do not improve PEFT models when selected randomly, however, we do see improvements in PEFT performance when combined with CED for selecting in-context demonstrations during both training and inference time. We also utilize the ability of a PEFT model, T-Few (Liu et al., 2022a), to train on very few examples to be able to effectively compute CED scores without overfitting to the target domain.

## 3 Methodology

We propose a method for selecting in-context demonstrations (ICDs) by finding the training data that would minimize the perplexity of a test example, if a language model were finetuned on that training example. This approach stems from previous findings that in-context examples may act as a type of meta-gradient on the frozen LLM (Dai et al., 2022) and the assumption that models perform better on in-domain test data. As we show in the following sections, our method of using cross entropy difference finds the demonstrations that appear most likely to be from the same domain as the test example.

### 3.1 Cross-Entropy Difference for In-Context Demonstration Selection

Generally, large language models are trained to minimize the negative log likelihood of a token given some context $\mathcal{C}$ by training on samples from a dataset $\mathcal{D}$. The parameters of the model are rep-

resented by $\theta$.

$$\mathcal{L}(\theta; \mathcal{D}, \mathcal{C}) = -\frac{1}{|\mathcal{D}|} \sum_{y \in \mathcal{D}} log P(y|\theta, \mathcal{C}) \quad (1)$$

In-context learning is the setting where the model weights $\theta$ are frozen and the only way to minimize the loss is to select an optimal context. In this work we also constrain the context to be examples from the training data. Note that the more general case is often referred to as prompt learning or prompt engineering where the context can include any natural language including instructions or descriptions in addition to examples.

We define a class of selection methods $\mathbb{W}$, were each function in the set outputs a subset of the training dataset and the size of the subset is at most $k$, where $k$ is the number of shots to include in the context. The selection may condition on the input so it is not restricted to selecting a single in-context demonstration for all test examples. The optimal selection method $\mathcal{W}*$ is defined as the selection method that minimizes the loss on the test domain.

$$\mathcal{W}^* = \arg \min_{\mathcal{W} \in \mathbb{W}} -\frac{1}{|\mathcal{D}|} \sum_{(x,y) \in \mathcal{D}} log P(y|\theta, \mathcal{W}(x))$$
$$(2)$$

Given a training set $D_{train} = \{(x_1, y_1), (x_2, y_2), ..., (x_n, y_n)\}$ with $n$ examples, where $x_i$ is the $i$-th input and $y_i$ is the corresponding label, the goal of few-shot learning is to learn a function $f : X \rightarrow Y$ that can accurately predict the label $y$ for a new input $x$, given only a small number of training examples. For simplicity of analysis, we focus on the case were only 1 demonstration is selected. This is especially useful for scenarios where each example is long, such as background documents. We leave multiple-demonstration selection to future investigation.

Cross-entropy difference (CED) is the difference between the cross-entropy losses from two models; a generic or base domain model (e.g. a pretrained language model) and a target domain model (e.g. a language model finetuned on a target distribution).

$$CED = log P(y|x; \theta_{target_i}) - log P(y|x; \theta_{base})$$
$$(3)$$

CED, in various forms, has been used extensively for data selection for domain adaptation (Axelrod et al., 2011; Moore and Lewis, 2010; Iter and

Grangier, 2021; Mindermann et al., 2022). Since we are selecting the argmax domain model per test example, as seen in Equation 2, in practice we can compare the cross-entropy of the target domain model directly as the base model loss is fixed per test example. Note that this differs from the standard data selection setting where there is one target domain and many data points which are scored.

For each $x_i$ in $D_{train}$, a separate target model is trained on the language modeling objective, producing $n$ models, $\mathcal{M}_1, ..., \mathcal{M}_n$. Given a test example $x_T$, we apply each $\mathcal{M}_i$ to compute $x_T$'s perplexity $L(\mathcal{M}_i(x_T))$. We then select the training sample associated with the language model giving the lowest perplexity as the in-context demonstration for $x_T$:

$$ICD = x_{\arg \min_i L(\mathcal{M}_i(x_T))} \quad (4)$$

Unlike the domain adaptation setting, rather than scoring all the training data using a single in-domain model, each training example is treated as its own domain. Each test example can be scored for "in-domain-ness" across all training examples.

To train each model on a single example, we use the (IA)$^3$ PEFT method with a T-Few 3B parameter model (Liu et al., 2022a). The effectiveness of PEFT to train a model on a small dataset without catastrophic forgetting allows us to train a model on a single example. The model is trained for multiple epochs and a small development set is used to test for overfitting and early stopping. Also, since a small fraction of parameters are updated, storing each model only requires 2MB on disk. Liu et al. (2022a) also showed that training a T-Few model was equivalent in FLOPs to approximately 20 forward passes of a 175B parameter GPT model. Since our finetuned models only require an average of 20 steps to train, the training cost in FLOPs per model is less than one forward pass of a 175B parameter GPT model.

### 3.2 Clustering for Large Training Sets

Training many small models on each training example and storing the respective weights may become prohibitively expensive, even when using small models trained with PEFT. We present a simple method to apply cross-entropy difference for in-context demonstration selection on larger training sets by clustering the training data. The approach from Section 3.1 can be scaled to larger training datasets by training each model $\mathcal{M}_i$ on a set of examples that can be obtained by clustering the

training data. Our experiments found that trivial clusters based on document embeddings did not perform well. Instead, we present a method for clustering which initializes $k$ finetuned models on $k$ random examples and uses cross-entropy difference to cluster the training data.

We sample a number of examples that is smaller than the size of the training datasets. For each example, a small model is trained with PEFT as in Section 3.1. All examples in the training dataset are scored with each model, producing a score for each example-model pair. The lower the loss on a model, the "closer" that point is to the model cluster. We use these scores to create equal sized clusters using only the initialization step for same-size k-means variation described by Schubert (2022). $\mathcal{M}_i$ models are retrained on each cluster, producing a new set of $k$ models. For selecting in-context demonstrations, we can either randomly sample from the cluster or use the original seed of the cluster (i.e. the centroid by CED).

## 3.3 ICDs as Meta-Gradients

Dai et al. (2022) describe in-context demonstrations as "implicit finetuning", where a component of the attention mechanism can be interpreted as a "meta-gradient". This formulation suggests that training directly on the in-context demonstration would have a similar effect to in-context learning with an LLM. Under this interpretation, the best selection strategy would be to choose examples that, if the model were to be trained on these examples, would result in the lowest loss on the test example. This strategy of decreasing the loss of a test example to measure domain similarity has been shown to correlate with performance on the downstream tasks in a domain adaptation setting (Grangier and Iter, 2022; Axelrod et al., 2011; Moore and Lewis, 2010). We apply the principle to the problem of in-context demonstration.

Dai et al. (2022) define an approximation to the standard attention head $Attn(V, K, q)$ as linear attention that removes the softmax and scaling factor. $V$, $K$, and $q$ are the value, key and query respectively and correspond to attention weight matrices, $W_V$ and $W_K$. We omit the full derivation from Dai et al. (2022) but include the expanded form of the linear attention in line 2 of Equation 5. $q$ is the attention query vector, $q = W_Q x$, and the input is $[X; X']$ where $X'$ is the in-context demonstration that is concatenated to the input $X$. Dai et al.

(2022) rewrites the linear attention head weight matrix as a reparameterization of the zero shot attention head $W_{ZSL}$ where the delta applied to the weights depends on the original wieghts and the in-context demonstration.

$$
\begin{aligned}
&Attn(V, K, q) \\
&\approx W_V X (W_K X)^T q + W_V X' (W_K X')^T q \quad (5) \\
&= (W_{ZSL} + \Delta W_{ICL}) q
\end{aligned}
$$

$\Delta W_{ICL}$ can be seen as an update that is applied to the zero shot weights of the attention mechanism $W_{ZSL}$. Here we see that including in-context demonstrations is akin to finetuning the LLM on the selected demonstrations.

If the loss of the large language model can only be modified by selecting in-context examples and these examples act as a meta-gradient on the language model, the optimal selection would be the training example with a gradient most similar to test example. Computing similarities between gradients would be computationally expensive given the size of large language models and gradients of a test example can not be computed because the model can not access the labels of test instances. Cross entropy difference (Axelrod et al., 2011), used in data selection for domain adaptation, has been show to be effective at selecting in-domain examples using the perplexity of the input features without the label.

Grangier (2019); Wang et al. (2020) describe cross entropy difference as an approximation of the dot product between the gradient of the target domain and the gradient of a single training example.

$$
\begin{aligned}
&logP(y|x; \theta_{target}) - logP(y|x; \theta_{base})_2 \\
&\approx \lambda g(x, y; \theta_{base})^T g(\mathcal{D}_{target}, \theta_{base})
\end{aligned} \quad (6)
$$

Here the cross entropy difference is approximating the gradient alignment between a single training example and a target domain. CED is simply defined as the difference between the log probabilities of a text span $y$ evaluated on two models, a base model and a target specific model. The base model represents the background distribution for which we can use any pretrained language model. The target model represents the distribution of a

---

[2]This holds when the base model and finetuned model are close, which is the case in finetuning. For ICDs and finetuning on a single example, this is even more limited than general finetuning.

target domain. Unlike cross entropy difference, in-context learning is input specific rather than dataset specific. To adapt the CED method to in-context demonstration selection, we need a model that is finetuned on a single example. In Section 3.1 we describe how we are able to finetune such a model with parameter efficient finetuning (PEFT) without overfitting to the single example and limiting the space requirements to store independent parameters per training example.

Equations 2 and 5 say that we want to find the examples that would minimize the loss if used as finetuning data. Equation 6 states that examples that have a gradient most similar to the actual test data can be approximated by finding the examples that most increase the likelihood of a test example. This provides the motivation for using CED to select ICDs. In the next section we describe in depth how to train models for each single-training-example domain and score the training data for selecting in-context demonstrations.

## 4 Experiments

We evaluate CED-ICD selection on both small models and the transfer of the selection method to larger models, including GPT-Davinci-003. We evaluate the selection method in a mixed-domain setting where random demonstrations are not trivially in-domain. We do not provide task or dataset labels as input to the selection model. We show in Section 5.2, both CED and the nearest neighbors baseline do not exclusively select in-domain demonstrations in the mixed domain setting. In fact, we show in Section 5.2 that out-of-domain examples may also be strong in-context demonstrations. In practical settings, a single LLM may be used for multiple tasks and there may not be labels for the task type, especially with the more common chat interfaces. We find this mixed-domain setting to better reflect these realistic challenges.

### 4.1 Datasets and Models

To evaluate the ICD-CED selection method, we measure the performance of several data selection methods on 8 datasets from 4 tasks; binary classification(BoolQ, Clark et al. (2019), NPBoolQ, Khashabi et al. (2020a)), extractive question answering (Squad2, Rajpurkar et al. (2018), NewsQA (Zhang et al., 2020)), abstractive question answering (NarrativeQA, Kočiský et al. (2018), NaturalQA, Kwiatkowski et al. (2019)) and multiple

choice (RACE, Lai et al. (2017), OpenBookQA Mihaylov et al. (2018)). All tasks are cast as a text generation task, in accordance to the evaluation used in UnifiedQA (Khashabi et al., 2020b). Binary classification and multiple choice are measured by accuracy, extractive QA is measured by F1 score based on the generated tokens and abstractive QA is measured by RougeL.

We combine these 8 datasets to create a larger mixed-domain dataset. We sample 32 examples from each dataset to create a medium-sized few-shot dataset with total of 256 training examples. We evaluate each dataset independently but with in-context demonstrations that can be selected from any dataset. We select one in-context demonstration as many examples have long "background" documents, where the input exceeds input length limits and need to be truncated.

We evaluate 2 settings, (1) small model performance combining PEFT with in-context learning and (2) in-context learning on LLMs. Our smaller model is T-Few 3B model (Liu et al., 2022a). Previous results don't report ICL performance because the authors did not find improvements from including ICL examples, however, as we show in our empirical results, T-Few can benefit from in-context learning if high quality demonstrations are selected. Further improvements are realized by finetuning T-Few with selected ICDs instead of random. For LLMs, we evaluate 3 sizes of GPT-3 (Babbage, Curie and Davinci (davinci-003)) (Ouyang et al., 2022).

We evaluate the following model settings, with the name corresponding to the rows in Table 1.
**Small Model Experiments**
*T-Few PEFT* is the standard parameter efficient finetuning setting where the model is finetuned on all the training data and inference does not include in-context demonstrations.
*T-Few PEFT + Random ICL* includes randomly selected in-context demonstrations at inference.
*T-Few PEFT + NN* uses OpenICL (Wu et al., 2023) to retrieve the most similar examples as measured by nearest neighbors in euclidean distance between embedding vectors.
*T-Few PEFT + CED* is our proposed model which selects in-context demonstrations using CED scores.
*T-Few PEFT + CED (training)* is our proposed model but includes using in-context selection during both training and inference.

| | Boolq (Acc) | NP-Boolq (Acc) | NarQA (RougeL) | NatQA (RougeL) | Squad2 (F1) | NewsQA (F1) | RACE (Acc) | OBQA (Acc) | AVG |
|---|---|---|---|---|---|---|---|---|---|
| Zero shot | 0.313 | 0.402 | 0.328 | 0.336 | 0.328 | 0.378 | 0.048 | 0.054 | 0.27 |
| PEFT FT | **0.629** | 0.574 | 0.347 | **0.376** | 0.368 | **0.509** | 0.552 | **0.576** | 0.49 |
| + Random ICL | 0.481 | 0.574 | 0.480 | 0.326 | 0.322 | 0.490 | 0.585 | 0.432 | 0.46 |
| + NN | 0.597 | 0.609 | **0.514** | 0.255 | 0.386 | 0.477 | **0.594** | 0.468 | 0.49 |
| + CED ICL | 0.621 | 0.602 | 0.506 | 0.265 | **0.414** | 0.471 | 0.580 | 0.510 | **0.50** |
| + CED ICL w/ (tr) | **0.629** | **0.695** | 0.504 | 0.276 | 0.399 | 0.467 | 0.593 | 0.469 | **0.50** |
| Loss Oracle | 0.875 | 1.000 | 0.515 | 0.392 | 0.703 | 0.767 | 0.750 | 0.781 | 0.72 |
| Oracle ICL | 0.969 | 1.000 | 0.650 | 0.454 | 0.784 | 0.794 | 0.813 | 0.906 | 0.80 |

Table 1: T-Few results on 8 datasets. Metrics are denoted next to dataset name.

| | Boolq | NP-Boolq | NarQA | NatQA | Squad2 | NewsQA | RACE | OBQA | AVG |
|---|---|---|---|---|---|---|---|---|---|
| Davinci Rand | 0.813 | 0.750 | 0.582 | 0.365 | 0.371 | 0.507 | 0.710 | 0.728 | 0.603 |
| Davinci NN | **0.860** | 0.751 | 0.549 | 0.379 | 0.366 | 0.532 | 0.780 | 0.813 | 0.629 |
| Davinci CED | 0.848 | **0.758** | **0.623** | **0.387** | **0.415** | **0.541** | **0.788** | **0.825** | **0.648** |
| + Clusters | 0.876 | 0.762 | 0.618 | 0.394 | 0.424 | 0.539 | 0.780 | 0.768 | 0.645 |
| Curie Rand | 0.662 | 0.648 | 0.387 | 0.253 | 0.259 | 0.405 | 0.228 | 0.306 | 0.394 |
| Curie NN | 0.678 | 0.634 | 0.396 | 0.264 | 0.297 | 0.386 | 0.292 | 0.336 | 0.410 |
| Curie CED | 0.776 | 0.678 | 0.407 | 0.294 | 0.392 | 0.430 | 0.344 | 0.356 | **0.460** |
| Babbage Rand | 0.542 | 0.482 | 0.316 | 0.220 | 0.213 | 0.369 | 0.254 | 0.306 | 0.338 |
| Babbage NN | 0.590 | 0.468 | 0.315 | 0.220 | 0.225 | 0.341 | 0.248 | 0.304 | 0.339 |
| Babbage CED | 0.664 | 0.504 | 0.321 | 0.242 | 0.281 | 0.371 | 0.262 | 0.322 | **0.371** |

Table 2: GPT-3 results using random, nearest neighbor and CDS in-context demonstration selection.

*T-Few PEFT + Loss Oracle* evaluates all available ICDs for each test example and reports the highest possible score when selected based on cross-entropy loss.

*T-Few PEFT + Oracle ICL* is similar to the above but the best ICD is selected based on the respective evaluation metric for each task as a true upper bound.

**Large Language Model Experiments**

*[GPT Model] Rand* randomly selects in-context examples similar to T-Few PEFT + ICL.

*[GPT Model] NN* uses OpenICL (Wu et al., 2023) to retrieve the most similar example from the test set as the in-context example.

*[GPT Model] CED* is our proposed model which selects in-context demonstrations using CED scores.

In-context demonstrations that do not fit entirely into the T-Few context window are truncated in the "background" section of the input exclusively, to keep the question, answer choices and answer intact. A simple prompt is used for GPT requests that labels the sections as "background", "question", "answer" and "example". We found that performance dramatically improved for binary classification by including an instruction to answer with a yes or no answer.

We also report the performance of CED-ICD with a larger set of candidate in-context demonstrations in Table 2, "+ Clusters" row. We sample 256 examples per task (total 2,048) and cluster them as described in Section 3.2. The total number of models used to select in-context demonstrations is 256, the same as the other rows in the table. The ICD for each cluster is selected as the cluster centroid (i.e. the seed example).

### 4.2 Results

Our results show that selecting in-context demonstrations using cross-entropy difference (CED) both outperforms baselines on a small trainable model and transfers to larger models, even improving results on GPT3-Davinci003. Table 1 reports the results of different selection methods on the T-Few 3 billion parameter model. Parameter efficient finetuning (PEFT) is a strong baseline that finetunes T-Few on the full set of training data, which is a total of 256 training examples. PEFT acheives the best results on T-Few for both BoolQ and NaturalQA datasets. Liu et al. (2022a) report that "[i]n preliminary experiments, we found that T0 was not able to perform few-shot ICL – performance actually decreased as we increased the number of in-context examples", which seems to be the case using random in-context demonstrations. However,

when incorporating stronger ICD selection methods, we show that performance does improve on NQ-BoolQ, NarrativeQA, Squad2, NewsQA and RACE. We found that T-Few does not perform well with in-context demonstrations if they are not included in the finetuning phase. When finetuning with in-context demonstrations, we evaluated both random ICD selection and CED selection. We found that on some datasets, we get further improvements by using CED selection during training as well as inference, which we expected as the training data will have more examples where the in-context demonstration is helpful for label prediction.

We report oracle in-context demonstration selection, the performance of an ideal example selector given the training data. In this setting, we evaluate every training example as an in-context demonstration and report the metric as the average of the best scores per example. Evaluating generated text requires a verbalizer to map the text to the labels for some tasks. Due to this mapping and metrics that do not directly correlate with loss, such as Rouge and token matching F1, we report both oracle scores based on selecting in-context demonstrations by the lowest loss and by the highest task metric performance. The latter is the true oracle performance but the former suggests that there may be some limitations to the extent that a cross-entropy based model may approximate the downstream performance on a task.

Oracle results show that there is still a large gap between our method and the oracle, showing that there may be many opportunities to improve smaller model performance with better selection methods. However, Figure 1 shows that very few examples yield substantial improvements over the average in-domain performance, meaning that while in-context demonstrations may come from a small set, statistical outliers may compound to make a significant improvement but a predictive method for determining which examples are outliers may not exist. Similar figures for all datasets are in the appendix.

Ultimately, in-context demonstrations are best used on large language models that can not be finetuned. Although our proposed selection method is based on the perplexities measured by smaller and finetuned models, we show that our selection method transfers to large models, including GPT-Davinci-003. These results are reported in Table 2.

Our proposed method for selection outperforms the baseline of using nearest neighbor retrieval on macro average across 8 datasets and on each of the 3 GPT model sizes evaluated. Clustering a larger set of candidate in-context demonstrations has similar performance overall. We use the same number of CED models (256) but train them on clusters of 2,048 candidate examples, showing we can maintain similar performance with more training data without substantially increasing compute. To estimate variance in this method, we sampled 256 training/in-context examples 5 times. We execute the end-to-end pipeline of training target models, selecting in-context demonstrations and running evaluation. From the evaluation sets, we bootstrap 50,000 sets to estimate the standard deviation. We find standard deviation on the text-davinci-003 model to be 0.011 and on text-curie-001 to be 0.007.

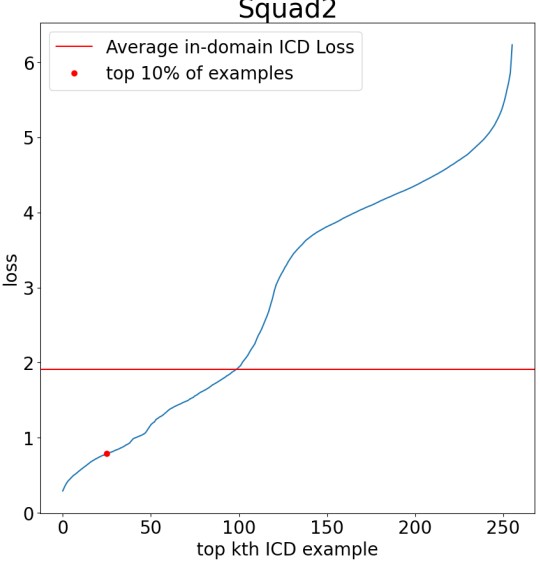

Figure 1: Losses for each in-context demonstration including both in-domain and out-of-domain examples for SQuAD2. Examples below the red line outperform the average in-domain performance.

## 5 Analysis

We analyze the quality of cross entropy difference selection by computing the rank of selected demonstrations compared to an oracle. We also explore the presence of strong in-context demonstrations selected from out-of-domain data, compared to in-domain data.

| Dataset | % In-Domain | | % In-Task | | Oracle | | |
|---|---|---|---|---|---|---|---|
| | NN | CED | NN | CED | In-Domain | In-Task | OOD |
| BoolQ | 0.50 | 0.50 | 0.72 | 1.00 | 0.91 | 0.91 | 1.00 |
| NP-Boolq | 0.84 | 1.00 | 0.84 | 1.00 | 0.88 | 0.91 | 1.00 |
| NarQA | 1.00 | 0.34 | 1.00 | 0.69 | 0.91 | 0.91 | 1.00 |
| NatQA | 0.66 | 0.91 | 0.69 | 0.91 | 0.84 | 0.91 | 1.00 |
| Squad2 | 0.34 | 0.47 | 0.63 | 0.78 | 0.91 | 0.97 | 0.94 |
| NewsQA | 0.56 | 0.81 | 0.63 | 0.81 | 0.97 | 0.97 | 0.94 |
| RACE | 0.75 | 0.84 | 0.78 | 1.00 | 0.94 | 0.94 | 0.97 |
| OBQA | 0.94 | 0.94 | 0.97 | 1.00 | 0.91 | 0.94 | 0.97 |
| Avg | 0.70 | 0.73 | 0.78 | 0.90 | 0.91 | 0.93 | 0.98 |

Table 3: The percentage of selected in-context demonstrations that are in-domain to the inference task is reported on the left. On the right, we report the percentage of oracle best in-context demonstrations that appear in each category, of in-domain, in-task and out-of-domain. Different in-context demonstrations can result in the same output and different outputs may score the same on the final metric so the oracle best in-context demonstration may appear both in-domain and out-of-domain.

| Dataset | NN | CED |
|---|---|---|
| Boolq | 24 | **8** |
| NP-Boolq | 23 | **4** |
| NarQA | **26** | 32 |
| NatQA | 20 | **12** |
| Squad2 | **10** | 23 |
| NewsQA | 31 | **24** |
| RACE | 15 | 15 |
| OBQA | 15 | 15 |
| Average | 20 | **16** |

Table 4: Average rank of the top 1 selected in-context demonstration for nearest neighbor selection and cross entropy difference selection. Rank is computed as the position is the full rank against all other in-context examples, computed using an oracle evaluated on the final metric for each dataset.

## 5.1 Ranking Selected ICDs

Oracle experiments provide a full ranking of all training data as in-context demonstrations. Table 4 shows the average rank of the top 1 selected in-context demonstration per dataset and average, comparing between CED selection and nearest neighbor selection. CED is better at selecting in-context demonstrations as a measured by the oracle ranking, 0 is the highest rank and 255 is lowest. The average rank of a CED selected demonstration is 16 out of 256.

Table 5 shows one example of in-context demonstrations selected by nearest neighbor and cross-entropy difference. While the nearest neighbor demonstration has significant lexical overlap with the test example, the demonstration is taken from a

different task. Conversely, the cross-entropy difference selected-demonstration is sampled from the same task and displays more similarity to the test example in the type of question.

## 5.2 In-Domain vs Out-of-Domain ICDs

Table 3 reports the percentage of selected in-context demonstrations that are from the same domain or same task as the inference dataset. The task refers to boolean classification, multiple choice, abstractive question answering and extractive question answering. Although the nearest neighbors approach attempts to directly find the most similar examples, cross entropy difference tends to select more in-domain demonstrations. In-domain demonstrations are a subset of in-task demonstrations so in-task selection percentage is always larger than in-task but CED has a larger proportion of in-domain selections indicating that CED is better at distinguishing the domain format even when there are other datasets that have a similar format or task structure.

Table 3 reports the percentage of oracle best in-context demonstrations that appear in each subset of in-domain, in-task and out-of-domain demonstrations. Different in-context demonstrations can result in the same output and different outputs may score the same on the final metric so the oracle best in-context demonstration may appear both in-domain and out-of-domain. Interestingly, this table shows that an oracle selection method that only has access to out-of-domain demonstrations can still achieve the best performance from this model on 98% of examples, showing that out-of-

| Question | Is the Tour de France different every year? |
| --- | --- |
| | *Context:* (Tour de France) Traditionally, the race is held ... circuits of France. |
| | *Answer:* Yes |
| NN-Selected In-context Demo | What is the name for the state secondary schools begun by Napoleon that were intended to standardize education across France? |
| | *Context:* (Napoleon) Napoleon's educational reforms laid the foundation ... outperformed its European counterparts, many of which borrowed from the French system |
| | *Answer:* Lycées |
| CED-Selected In-context Demo | Is china the third largest country in population? |
| | *Context:* (China) China, officially the People's Republic of China (PRC), is ... and the special administrative regions of Hong Kong and Macau. |
| | *Answer:* No |

Table 5: An example of an in-context example selected by nearest neighbor versus one selected by Cross-Entropy Difference.

domain selection of in-context demonstrations can be highly effective. This suggests that for datasets that have very few or no demonstrations, out-of-domain demonstrations may still improve model performance.

# 6 Conclusion

This work shows that cross entropy difference can be used as a heuristic for selecting in-context demonstrations for in-context learning. We motivate our approach by linking previous observations that in-context demonstrations operate as meta-gradients on a frozen large language model to the data selection literature which has shown that CED is a method that selects examples that have a gradient most similar to the in-domain examples. We empirically show that we can use smaller models to compute CED scores and that this selection method effectively transfers to large models, such as the 175 billion parameter GPT3, and is able to improve performance over baseline selection methods.

# 7 Limitations

The main limitation of this work is the added complexity of training multiple small models and the tradeoff between this extra computational and space cost and the improvement over baselines that may require less computation. While PEFT training tends to be stable, it does require searching for an appropriate learning rate, although in this work we found 3e-2 to be appropriate for all datasets. As we report, clustering and using fewer models does degrade performance so a valuable direction for future work would be to better scale to larger training datasets. Also, as mentioned earlier, we

focus on the one shot in-context learning setting as many of the datasets that we evaluate contain "background" as part of the input such that including one in-context demonstration often exceeds the maximum input length for both GPT and T-Few. This requires careful truncation that may effect performance. However, recent GPT-4 releases have included extended input lengths, up to 32,000 tokens, which allow for many more in-context examples and overall better performance on tasks.

Although we evaluate difference sizes of LLMs, we only evaluate using GPT models. With recent releases of open-source LLMs such as LLaMa (Touvron et al., 2023), it would be of value to compare the effectiveness of different selection methods across different flavors of LLMs. Furthermore, while there are significant limitations for finetuning or accessing LLM weights of proprietary GPT models, open-source models open the opportunity to include a finetuning phase for the LLM or using activations from the LLM as a signal for the selection method.

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

## 8 Appendix

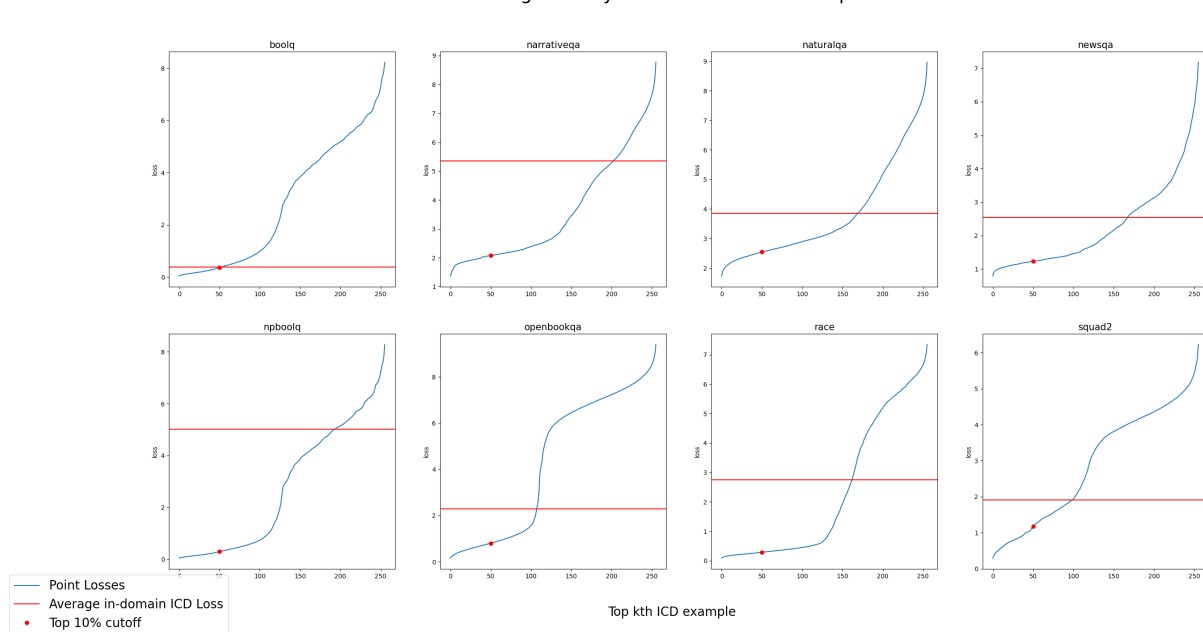

Figure 2: Losses for each in-context demonstration including both in-domain and out-of-domain examples for all datasets. Examples below the red line outperform the average in-domain performance.