# OpenReview forum: "In-Context Demonstration Selection with Cross Entropy Difference"
_EMNLP/2023/Conference — EMNLP 2023 Findings_

### Official Review · Reviewer_EXPc · 2023-07-25

**Soundness:** 4

**Excitement:**

4: Strong: This paper deepens the understanding of some phenomenon or lowers the barriers to an existing research direction.

**Paper Topic And Main Contributions:**

This paper proposes a method to efficiently select in-context examples from a subset of the training data that minimize the loss of the test example. The gist of the method is based on the cross-entropy difference in losses between the target domain model and a base domain model, and it builds on existing solid theory in this research area. The paper is well written, has extensive experiments and high-quality baselines, relevant results demonstrating the benefit of the method, especially for larger LLMs where fine-tuning might have prohibitive cost.

**Questions For The Authors:**

* Can you quantify the cost of training M models, and how this cost depends on the size of M?

* What happens to the cost as well as evaluation metrics if we scale number of training examples beyond 256?

* Is it a fair interpretation that you actually don't use CE *difference* but only the cross entropy of the target model? I might have gotten confused here, but would appreciate some explanation here.

* How robust are your results to different subsets of 256 training examples? (The real answer here that would make me very happy would be to bootstrap different subsets and report some measure of confidence on this)

* L479: "We found that performance dramatically improved..." --> did you also find this for T-Few?


**Reasons To Accept:**

- The proposed method appears to work well
- Robust evaluation, good theoretical grounding in existing work, extensive coverage of experimental benchmarks and different baselines, validation of results
- Strong contextualization of research in existing, well established domain in meta-gradient learning in LLMs
- Evaluation across both small and large LLMs, with many different experimental conditions



**Reasons To Reject:**

- Focus on a single demonstration - commentary on how this might impact performance in tasks that do not have long background contexts is welcome
- Potential issues with scalability as the training set size is limited to 256, and it is not clear how robust the method is to a selection of that subset

**Reproducibility:**

4: Could mostly reproduce the results, but there may be some variation because of sample variance or minor variations in their interpretation of the protocol or method.

**Reviewer Confidence:**

4: Quite sure. I tried to check the important points carefully. It's unlikely, though conceivable, that I missed something that should affect my ratings.

**Typos Grammar Style And Presentation Improvements:**

First, thank you for generally well written paper that makes my job as a reviewer much more pleasant. Second, there are some minor issues with formatting and typos, here are a few:

* L274: All examples in the training dataset are scores with each model -> are scored?
* L287: Dai et al. (2022) describes
* L304: (Dai et al. 2022) define
* Footnote 1: "This hold"
* L367: Equations 6 states

---

> ### Author Rebuttal · Authors · 2023-08-25
>
> Thank you for your review and suggestions. In particular, thank you for your close reading, formatting/ typo suggestions and thoughtful questions!
> We address your questions here:
> 	• Each model is trained on a single V100 GPU and only requires around 20 steps (we use early stopping). Training time is a few minutes including validation iterations. Each model is independent so this training is done M times. For larger training datasets, we present a method for clustering the data such that M can be fixed as the training set grows.
> 	• There are diminishing returns as the size of the training data grows. Increasing the number of training examples leads to increasing the cost of training the M selection scoring models and one additional forward pass during inference per test example. Comparing these costs to a single inference on a 175B GPT model, training a T-Few PEFT model and doing inference on that model is fewer FLOPs than one inference with 175B GPT. Liu et al (https://arxiv.org/pdf/2205.05638.pdf) does this analysis in Table 1. Note that we only train for 20 steps instead of 1000 for the full T-few model so our training cost would be about 7.7e14 x M based on these estimations. These costs are still much lower than the cost to run the full evaluation with GPT-Davinci.
> 	• Yes, it is correct that while the derivation for this method is based on CE difference, since the base model is the same for each target model we can simplify by just using cross entropy. However, if we had used different base models (for example, if we used a medical or finance domain pretrained model if we knew some subset of training data were from those domains) then we would need to compute the difference.
> 	• Thank you for the suggestion! While variance can be large for in-context learning per example, the overall performance per evaluation set is more robust. We will add confidence intervals to the camera-ready if accepted.
> No, we did not find adding instructions to T-Few to help. In our experiments, we found smaller models such as T-few 3B to not have the instruction following capabilities of GPT.

---

### Official Review · Reviewer_uof7 · 2023-07-30

**Typos Grammar Style And Presentation Improvements:** 056
**Soundness:** 3

**Excitement:**

3: Ambivalent: It has merits (e.g., it reports state-of-the-art results, the idea is nice), but there are key weaknesses (e.g., it describes incremental work), and it can significantly benefit from another round of revision. However, I won't object to accepting it if my co-reviewers champion it.

**Paper Topic And Main Contributions:**

This paper proposes a method for selecting good examples as demonstrations for in-context learning. The proposed method is inspired by the cross-entropy method in the literature commonly used for data selection for domain adaptation. An adaptation of the method for demonstration selection is proposed by the authors. For each training example, a separate model is trained (using parameter efficient tuning techniques) and the best example is chosen based on the fine-tuned model that has lowest test perplexity on the test query. A clustering based approach is also considered which trains models on groups of examples instead of individual examples. Results across 8 benchmarks show that the proposed method helps improve the performance over other strategies for choosing demonstrations based on the T-Few and GPT models.

**Questions For The Authors:**

A: How do the performance numbers presented here compare to published numbers on these benchmarks? It would be more impressive if the proposed method can be used to improve performance over published numbers.

B: The paper mentions that they only focus on the setting where 1 demonstration is selected. This seems like a very limited setting. Perhaps the top-k demonstration chosen based on the proposed strategy can be concatenated and provided as a prompt?

C: How many validation examples were used for training the models?

D: If 32 examples from each dataset is available, why is it that only 1 example is included as a demonstration? Is the information in the remaining examples neglected? I understand that some datasets only allow 1 example to be used due to length issues, but that doesn’t limit us from using more examples when they do fit the length constraints. I also find the setting where examples from all tasks are combined together somewhat less natural. This makes it a multi-task learning setting and I feel this may put the PEFT baseline at a disadvantage.

E: Which PEFT method is used for training? What is the impact of the specific PEFT method on the final performance? This analysis needs to be presented in an ablation.

F: I find it intriguing that when the same model used for example selection is used for in-context learning as well, the performance is not consistently better (e.g., Table 1). However, when a different model is used for in-context learning (GPT), the results are better. What is the explanation for this behavior? Also, how does the theory work in this context (e.g., when different models are used for example selection and inference)?

G: How important is the choice of the example selection model (e.g., T-Few)? What happens when different models are used instead?

H: Apart from OpenICL, were any other retrieval mechanisms attempted? It would also be helpful to show qualitative examples of retrievals from proposed and baseline methods.

**Reasons To Accept:**

* Given the popularity and convenience of in-context learning, exploring strategies for efficiently utilizing this capability of language models is important.
* Proposed method helps improve performance over baseline strategies for choosing demonstrations.
* Proposed method is justified with theoretical connections.

**Reasons To Reject:**

* The results are generally mixed, the proposed method does not always consistently improve performance (e.g., Table 1).
* Some experimental design choices are not well justified such as combining datasets from multiple tasks and restricting number of demonstrations to 1 example. Some ablations are missing that analyze the impact of the choice of PEFT method, model used for example selection, choice of retriever used as baseline. This makes it hard to gauge the impact of the method. See Questions section for details.
* Technical contribution seems limited as the method is largely an application of an existing method developed for example selection in domain adaptation.

**Reproducibility:**

3: Could reproduce the results with some difficulty. The settings of parameters are underspecified or subjectively determined; the training/evaluation data are not widely available.

**Reviewer Confidence:**

3: Pretty sure, but there's a chance I missed something. Although I have a good feel for this area in general, I did not carefully check the paper's details, e.g., the math, experimental design, or novelty.

---

> ### Author Rebuttal · Authors · 2023-08-25
>
> Thank you for your review and your thoughtful questions and suggestions for improving the paper. We will make the suggested corrections in the camera ready if accepted and we address each question below.
> 	A. We use the described dataset to evaluation and analyze the proposed method on a cross-domain multi-task few shot dataset. We are not aware of previously published benchmarks on such a dataset. There are single domain few-shot datasets that we hope to evaluation in future work.
> 	B. The inputs for the tasks that we evaluate are long. For the majority of examples, only 1 in-context demonstration fits entirely. Adding more would require non-trivial truncations and we found that naively packing other examples with naïve truncation led to worse performance. However, since the method produces scores, we can trivially rank all demonstrations and use top-k in future explorations.
> 	C. Since we have 32 examples per dataset and we train a PEFT model on 1 example, we can use the other 31 as validation. In practice, 4-8 are sufficient to detect an acceptable stopping point.
> 	D. In many real world settings, LLMs are used for many tasks at the same time, such as a chatbot or assistant. Task performance can be improved by finding in-context examples but what domain to select demonstrations from is unknown. Furthermore, as we show in this work, sometimes demonstrations sampled from other domains may be as good or better than sampling from a finite in-domain set. Our evaluation setting is meant to evaluate and analyze such a setting on conventional academic datasets.
> 	E. Our PEFT method is IA3 (https://arxiv.org/pdf/2205.05638.pdf). We agree that evaluating different PEFT methods for this task would be valuable for the community and may add an ablation in the camera-ready if accepted.
> 	F. Smaller models (such as 3B T-Few) are not as effective at using in-context demonstrations as larger GPT models. This is consistent with findings in the T-Few paper as well. Regarding the theory, the smaller model is used to estimate the "in-domain-ness" of an example so there is no constraint on similarity between selection and inference models, assuming both are generic-domain. However, you bring up a good point that this may not be as effective for specialized models, such as a medical domain model. This is an interesting idea for future exploration!
> 	G. Our unreported experiments found that the most important feature of the example selection model is that it does not overfit to the training data. For example, not using PEFT (ie. Full model finetuning) quickly overfit and the model was no longer effective for example selection.
> 	H. Nearest-neighbor and random were the most common baselines from literature and applications so we focused on these baselines. Thank you for the suggestion. We will add qualitative examples of retrievals to the camera-ready if accepted.
>
> Regarding you comment on the technical contributions being limited, we would like to point out that we are the first to explore cross-entropy difference in the context of large language models and in-context demonstration selection. Furthermore, our proposed approach of training on a model on the training data and scoring each test example with multiple training-data models is novel. Domain adaptation was an influence of this work, but in that setting, the selection model is trained on a separate in-domain set and the training data is scored for selecting top-k examples to train or finetune an "in-domain" model. I hope this clarifies the novelty of our proposed method. We will also try to clarify this in the description of the method in the paper.

---

### Official Review · Reviewer_YS4p · 2023-08-01

**Typos Grammar Style And Presentation Improvements:** Table 4 format is different from the …
**Soundness:** 3

**Excitement:**

3: Ambivalent: It has merits (e.g., it reports state-of-the-art results, the idea is nice), but there are key weaknesses (e.g., it describes incremental work), and it can significantly benefit from another round of revision. However, I won't object to accepting it if my co-reviewers champion it.

**Paper Topic And Main Contributions:**

This paper proposes a demonstration selection strategy based on cross-entropy over the target data, which first trains a base model on training data and then infers on test input to select the demonstration instances by calculating the cross entropy. Experimental results on both small model and LLMs settings demonstrate the effectiveness of the proposed demonstration selection method.

**Questions For The Authors:**

I am confused by the calculation of the CED, how to calculate the gradient of target input without knowing its label and its domain? How to choose the training data for PEFT? Could you please give more details?

**Reasons To Accept:**

The proposed demonstration selection strategy is novel and effective. Extensive experiments show that concatenating the demonstrations selected by the proposed method can lead to improved performance across multiple benchmarks.

**Reasons To Reject:**

The proposed method incurs much higher computational and storage costs compared to baseline methods. When dealing with a large number of tasks, the use of CED-based demonstration selection may not be a worthwhile approach, as the improvement is not significant and stable on some datasets.

**Reproducibility:**

2: Would be hard pressed to reproduce the results. The contribution depends on data that are simply not available outside the author's institution or consortium; not enough details are provided.

**Reviewer Confidence:**

3: Pretty sure, but there's a chance I missed something. Although I have a good feel for this area in general, I did not carefully check the paper's details, e.g., the math, experimental design, or novelty.

---

> ### Author Rebuttal · Authors · 2023-08-25
>
> Thank you for your review! To calculate cross entropy / gradients we use the language modeling objective. Therefore we do not need the label for any input. This is similar to computing the perplexity of a text. The training data for the PEFT models is the same as the training data used for in-context demonstration selection so the two settings are evaluated on exactly the same data.
>
> Regarding your comment about the computational and storage costs, it is true that there are extra costs to training the domain-specific models and storing their weights. However, we would like to point out that because we use PEFT it is only 4MB per model and if using the clustering approach that we present, the storage costs are fixed as we scale to more tasks. For the extra computational cost, to put into perspective the extra cost of training these smaller PEFT models, Table 1 in Liu et al (https://arxiv.org/pdf/2205.05638.pdf) shows that training T-Few is equivalent in FLOPs to approximately 20 requests to GPT-3 175B model so even the cost of running evaluation on the test set is much larger than the cost of training these extra models.
>
> Regarding reproducibility, all data is publicly available and we will be releasing our source code and exact data splits so experiments will be easily reproducible.
>
> Thank you again for taking the time to review our paper and your suggestions to improve our presentation.

---

### Meta-Review · Area_Chair_pSUV · 2023-09-15

**Recommendation:** 4

**Metareview:**

The paper presents a novel method, named cross-entropy difference (CED), for in-context demonstration selection in Large Language Models (LLMs). The method utilizes parameter efficient fine-tuning and focuses on improving performance on zero-shot tasks by selecting effective in-context demonstrations. The evaluation is done on a mixed-domain dataset that combines various benchmarks and tasks. The results show the proposed method outperforms baseline selection methods for a range of LLMs.

**Summary of Reviews and Author Rebuttal:**

Reviewer YS4p found merit in the paper's novel demonstration selection strategy. The paper's strengths include the novelty of the proposed strategy and the experimental validation that suggests performance improvements across various benchmarks. The criticisms were primarily around the computational and storage costs of the method. Additionally, the reviewer expressed confusion regarding the calculation of CED and had some concerns about the paper's reproducibility. In their rebuttal, the authors addressed the computational and storage concerns, citing the PEFT approach's efficiency and pointing out that costs scale favorably as tasks increase. The authors also clarified the method they used to calculate cross entropy. Furthermore, they responded to the reproducibility concerns by committing to release both their code and data splits, ensuring that their experiments would be replicable.

Reviewer uof7 acknowledges the relevance of the study given the rising popularity of in-context learning. The method showcased an improvement over the baseline strategies for choosing demonstrations and had theoretical justifications. However, the reviewer points out mixed results, especially in Table 1, and several experimental design choices that were not sufficiently justified. They also raised questions about the novelty of the method, as it seems to be an adaptation of an existing method. The author's rebuttal addressed most of the concerns, highlighting that their work introduces cross-entropy difference in the context of LLMs and in-context demonstration selection for the first time.

Reviewer EXPc provided a favorable assessment of the paper, praising its theoretical grounding, experimental coverage, and the robust evaluation. However, the reviewer also raised concerns about potential scalability issues and sought clarifications on the cost of training models, the effect of scaling training examples, and the method's robustness. In their rebuttal, the authors addressed each concern, explaining their methodology further and providing insights into how the training works with different dataset sizes. The author's responses appeared to satisfy the concerns, and the reviewer acknowledged the rebuttal positively.

**Soundness and Excitement:**

There seems to be a consensus that the method presented is solid, both in its theoretical grounding and empirical evaluation. The reviewers assigned a high score for soundness, noting the robustness and the extensive coverage of experimental benchmarks.

The paper contributes to the research domain of in-context learning in LLMs and offers a new perspective on demonstration selection. While the idea and the results are noteworthy, the trade-off between performance improvement and the computational/storage costs leads to average excitement around the paper.

---

### Decision · Program_Chairs · 2023-10-07

**Decision:**

Accept-Findings

**Comment:**

The paper presents a novel method, named cross-entropy difference (CED), for in-context demonstration selection in Large Language Models (LLMs). The method utilizes parameter efficient fine-tuning and focuses on improving performance on zero-shot tasks by selecting effective in-context demonstrations. The evaluation is done on a mixed-domain dataset that combines various benchmarks and tasks. The results show the proposed method outperforms baseline selection methods for a range of LLMs.

**Summary of Reviews and Author Rebuttal:**

Reviewer YS4p found merit in the paper's novel demonstration selection strategy. The paper's strengths include the novelty of the proposed strategy and the experimental validation that suggests performance improvements across various benchmarks. The criticisms were primarily around the computational and storage costs of the method. Additionally, the reviewer expressed confusion regarding the calculation of CED and had some concerns about the paper's reproducibility. In their rebuttal, the authors addressed the computational and storage concerns, citing the PEFT approach's efficiency and pointing out that costs scale favorably as tasks increase. The authors also clarified the method they used to calculate cross entropy. Furthermore, they responded to the reproducibility concerns by committing to release both their code and data splits, ensuring that their experiments would be replicable.

Reviewer uof7 acknowledges the relevance of the study given the rising popularity of in-context learning. The method showcased an improvement over the baseline strategies for choosing demonstrations and had theoretical justifications. However, the reviewer points out mixed results, especially in Table 1, and several experimental design choices that were not sufficiently justified. They also raised questions about the novelty of the method, as it seems to be an adaptation of an existing method. The author's rebuttal addressed most of the concerns, highlighting that their work introduces cross-entropy difference in the context of LLMs and in-context demonstration selection for the first time.

Reviewer EXPc provided a favorable assessment of the paper, praising its theoretical grounding, experimental coverage, and the robust evaluation. However, the reviewer also raised concerns about potential scalability issues and sought clarifications on the cost of training models, the effect of scaling training examples, and the method's robustness. In their rebuttal, the authors addressed each concern, explaining their methodology further and providing insights into how the training works with different dataset sizes. The author's responses appeared to satisfy the concerns, and the reviewer acknowledged the rebuttal positively.

**Soundness and Excitement:**

There seems to be a consensus that the method presented is solid, both in its theoretical grounding and empirical evaluation. The reviewers assigned a high score for soundness, noting the robustness and the extensive coverage of experimental benchmarks.

The paper contributes to the research domain of in-context learning in LLMs and offers a new perspective on demonstration selection. While the idea and the results are noteworthy, the trade-off between performance improvement and the computational/storage costs leads to average excitement around the paper.